# The internal and marginal adaptation of lithium disilicate endocrowns fabricated using intra and extraoral scanners: An in-vitro study

Marzieh Akhlaghian[1], Amir-Alireza Khaledi[1], Seyed Ali Mosaddad[2,3], Sana Dabiri[4], Rashin Giti[1]*, Farhad Kadkhodae[4], Shabnam Gholami[4]

1 Department of Prosthodontics, School of Dentistry, Shiraz University of Medical Sciences, Shiraz, Fars, Iran, 2 Department of Research Analytics, Saveetha Institute of Medical and Technical Sciences, Saveetha Dental College and Hospitals, Saveetha University, Chennai, India, 3 Faculty of Odontology, Department of Conservative Dentistry and Bucofacial Prosthesis, Complutense University of Madrid, Madrid, Spain, 4 Student Research Committee, School of Dentistry, Shiraz University of Medical Sciences, Shiraz, Iran

* giti_ra@sums.ac.ir

## Abstract

**Data Availability Statement:** All relevant data are within the manuscript and its Supporting information files.

### Objectives

The impression technique highly influences the adaptation of ceramic restorations. Not enough information is available to compare the marginal (MF) and internal fit (IF) of endo-crowns fabricated with various digitization techniques. Therefore, this in-vitro study aimed to compare the MF and IF of lithium disilicate (LDS) endocrowns fabricated through direct and indirect digital scanning methods.

### Materials and methods

One extracted maxillary molar was used to fabricate endocrowns. The digitization of the model was performed with (G1) direct scanning (n = 10) utilizing an intraoral scanner (IOS), (G2) indirectly scanning the conventional impression taken from the model using the same IOS (n = 10), (G3) indirectly digitalizing the obtained impression using an extraoral scanner (EOS) (n = 10), and (G4) scanning the poured cast using the same EOS (n = 10). The MF and IF of the endocrowns were measured using the replica method and a digital stereomicroscope. The Kruskal-Wallis test was used to analyze data.

### Results

The studied groups differed significantly (p<0.001). G2 (130.31±7.87 µm) and G3 (48.43 ±19.14 µm) showed the largest and smallest mean vertical marginal gap, respectively. G2 and G3 led to the highest and lowest internal gaps in all regions, respectively. With significant differences among the internal regions (p<0.001), the pulpal area demonstrated the most considerable misfit in all groups.

**Funding:** The author(s) received no specific funding for this work.

**Competing interests:** The authors have declared that no competing interests exist.

## Conclusions

Scanning the impression using an extraoral scanner showed smaller marginal and internal gaps.

## 1. Introduction

Current dentistry concepts are shifting toward more minimally invasive approaches, conserving as much tooth structure as possible [1]. Thus, conventional concepts of achieving resistance and retention in prosthetic designs are now being replaced by the development of adhesive dentistry principles [2], shedding light on the importance of preserving tooth biorim, bonding protocols, and biomimetic approaches [3]. In reconstructing a heavily destructed tooth structure, ceramic restorations have thus been developed to serve the purpose of conservative dentistry [4] by providing an intaglio surface that could be bonded to the remaining tooth structure [5]. One of the existing ceramic restorations is endocrown, a monolithic partial restoration covering the entire occlusal surface [6], which bonds to the prepared circumferential margins and the pulp cavity of a root canal-treated tooth through its intaglio surface [7]. The adhesive cementation and prepared tooth's pulpal walls provide micro- and macro-mechanical retention, respectively [8]. This restoration is highly esthetic and minimally invasive, with fewer fabrication requirements than conventional crowns [9]. Endocrowns are indicated in limited interocclusal space and for molars with severely curved, resorbed, short, or weak roots unsuitable for intraradicular post-fabrication [10]. One ceramic material in fabricating endocrowns is lithium disilicate (LDS). It is a glass-based ceramic that can be processed using either a heat-pressed technique or milled using digital techniques [11]. High fracture strength, modulus of elasticity, flexural strength, esthetic appeal, and outstanding biocompatibility features define LDS [12].

The proximity degree between the restoration (intaglio and marginal surfaces) and abutment tooth (surface and cavo-surface angles) [13] can be used to measure the internal (IF) and marginal fit (MF) of restoration [14]. Because an accurate intraoral impression is required for manufacturing a well-fitting prosthesis [15], the greater the impression accuracy, the closer the restoration-abutment proximity, and the greater the IF and MF [13]. The MF is the most crucial factor in establishing a prosthesis's long-term functional success rate [16]. The MF is affected by the preparation and marginal design, type of restorative material, impression approach used, and fabrication technique [17]. Marginal discrepancies increase cement dissolution, plaque deposition, periodontal inflammation, and recurrent caries [18], ultimately leading to restorative failure [19,20]. The probability of recurrent carries and prosthetic failure arises from a poor cement seal that allows microorganisms to enter [21]. A thick layer of cement compromises the fracture resistance of a ceramic restoration due to increased interfacial stress and polymerization shrinkage [22]. Furthermore, a poor IF could increase stresses at the tooth-restoration interface and decrease fracture resistance [23]. There is no agreement on the maximum allowable marginal (MG) and internal gap (IG) for clinical success; however, values ranging from 50 to 200 μm have been documented [16,18,24,25].

Adhesive dentistry has been revolutionized since the development of digital dentistry. Computer-aided design and computer-aided manufacturing (CAD/CAM) technology have rapidly become popular in dentistry, providing more time-efficient and accurate impressions with superior patient comfort and the benefit of real-time visualization [10] without the drawbacks of conventional methods, including material considerations, office-laboratory transportation, disinfection requirements, adherent inaccuracies with impression techniques, trays and

mixing approaches, low working time, patient distress, the disagreeable taste of the impression materials, and nausea [16]. Scanning accuracy is essential for the digital workflow to digitalize the prepared tooth surfaces precisely. A digital scan with high accuracy can improve the compatibility of the fabricated prostheses, IF, MF, and clinical success rates [10]. The first step in a digital process is either taking a direct digital intraoral impression using IOSs or indirectly digitizing a conventional impression/poured cast by IOSs or extraoral scanners (EOSs) [19]. However, the accuracy of a digital impression can be affected by finish line position, access difficulties, periodontal status, sulcular bleeding during scanning, salivary flow rate, patient compliance, the necessity for powder application for some devices, operator's skill, and environment lightening [19,26–28]. IOS requires numerous digital datasets to be acquired and merged due to its small scanning tips. In the digital scan and the final prosthesis, this matching process inevitably adds a minor systematic error [29]. Extraoral digitization is further affected by scanning errors resulting from dimensional distortion in the impression material and the produced gypsum cast [30] and integrated errors with the stone cast pouring and conventional impression taking [31]. Due to a lack of available literature comparing the adaptation of endocrowns made using various digital scanning approaches, this in-vitro study aimed to determine how direct and indirect scanning techniques using IOS and EOS could affect the MF and IF of LDS endocrown restorations. According to the null hypothesis, there would be no differences in the degree of marginal and internal discrepancies between restorations fabricated using different digital approaches.

## 2. Materials and methods

This in-vitro study was approved by the Ethical Principles and National Norms and Standards for Conducting Medical Research in Iran under identifiers IR.SUMS.DENTAL.REC.1400.075, 1401.113, and 1401.114.

### 2.1. Design and preparation of the tooth

One human permanent maxillary first molar, found hopeless by a periodontist due to severe periodontitis, was selected for this research. The tooth was extracted as atraumatically as possible. An ultrasonic device (Ultrawave™ XS, Ultradent Products, Inc., USA) was used to remove any remaining tissues, adhered plaque, or calculus. The tooth was inspected using a magnification loupe (3.5×) and light to ensure it was free from cracks, anatomical defects, and carious lesions. The specimen was immersed in 5.25% sodium hypochlorite (NaOCl) for 1 hour. To prevent dryness, it was then kept in normal saline until the endodontic treatment [32].

An endodontic access cavity was initially performed using a fissure diamond bur, and the pulp tissue was removed. The preparation of the pulp cavity was finalized using a round carbide bur. The crown-down method was applied to prepare the canals with rotary instruments. The root canals were finally obturated using a sealer (AH 26; Dentsply Sirona) and gutta-percha (Gutta Percha Points; Meta Biomed, South Korea) using the lateral condensation method.

The tooth was then embedded into a self-curing acrylic resin block (Ivocron, Ivoclar Vivadent AG) to a depth of 2 mm below the cemento-enamel junction (CEJ). A small tungsten carbide rotary instrument (Hager & Meisinger GmbH) was used to remove the gutta-percha from each root canal's initial 2 mm entrance. The canal orifice was then filled using a flowable composite resin (CLEARFIL MAJESTY ES Flow, Kuraray Medical, Okayama, Japan) [33]. According to Fig 1A, a diamond disk (918 BF, DZ, Lemgo, Germany) at a low speed was employed to section the tooth 2 mm above the CEJ and parallel to the occlusal surface to prepare a butt-joint design [34,35]. Using a periodontal probe suggested the preparation was performed at a 4–5 mm depth of the reduced walls, measured through the access cavity. To reach a slight

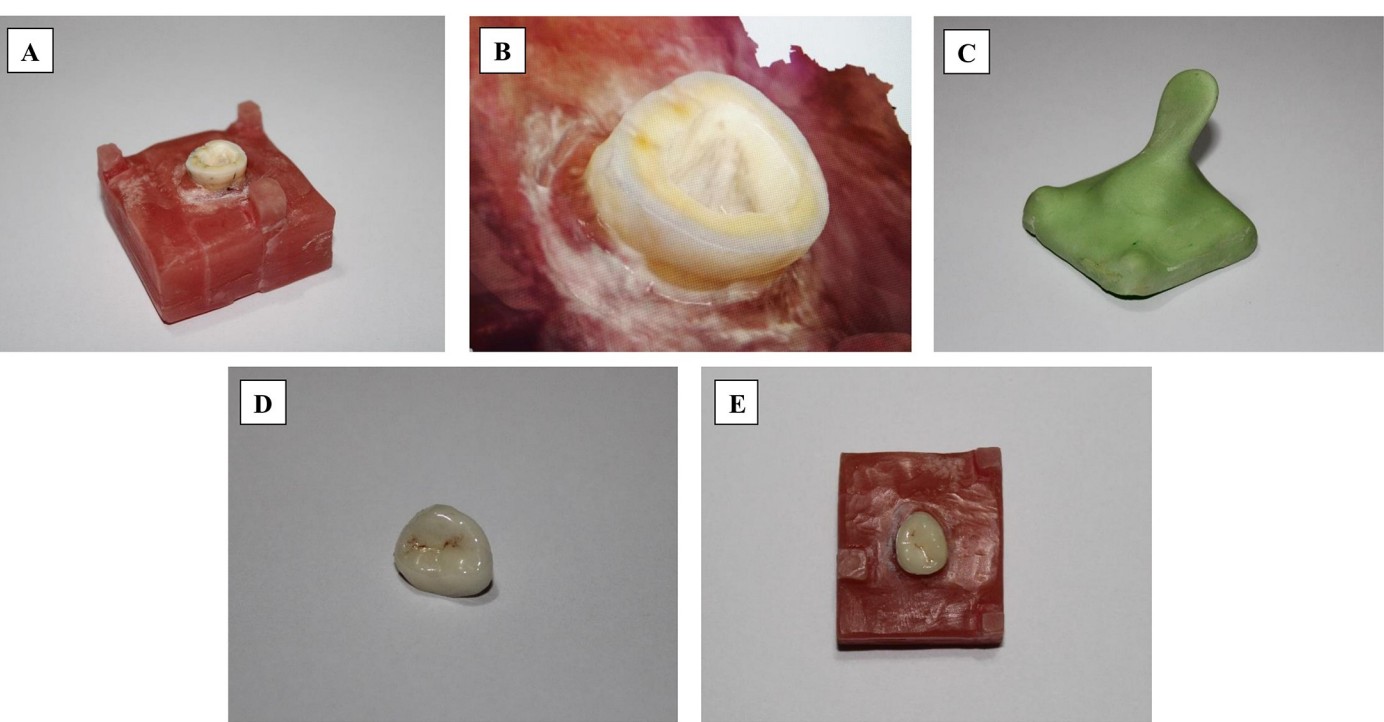

**Fig 1.** **(A)** The prepared and mounted tooth model for the fabrication of endocrown; **(B)** The scanned tooth via direct approach using an intraoral scanner; **(C)** The special tray for taking a conventional impression; **(D)** A fabricated and glazed endocrown; **(E)** The fabricated restoration seated on the tooth.

internal taper (8–10º), the axial walls were prepared using a round-end taper diamond bur (Hager & Meisinger GmbH). This instrument also helped flatten the pulp chamber base and remove excessive retentive areas and round internal angles [36]. Smooth internal transitions in the finishing procedure were performed using a bur (Drendel + Zweiling Diamant GmbH, Germany) with a larger diameter and a finer particle size [37]. Prepared margins were polished in the polishing phase using fine diamond burs and composite polishing instruments (EVE, Ernst Vetter GmbH, Germany). 95% ethanol was used to clean the pulpal cavity. Afterward, the prepared tooth model was digitalized using direct and indirect scanning methods based on the grouping.

The required sample size was estimated, employing data from previously published investigations [7,17]. Using a statistical power analysis program (G*Power v. 3.0.10, Heinrich-Heine-University Düsseldorf, Düsseldorf, Germany), the sample size was calculated to compare research groups while taking power (80%), α (0.05), and effect size (0.6) into account. The total number of samples was rounded to 40 (ten samples in each study group) from the calculated sample size of 36.

## 2.2. Group 1—Scanning the tooth with an intraoral scanner

A single operator (M.A.) made ten direct digital scans of the prepared tooth model using an intraoral scanner (3Shape TRIOS 3, Copenhagen, Denmark) (Fig 1B). One axial wall was the starting point for scanning, continued with the occlusal surface and ended at the opposing wall. The confocal laser mechanism-based high-speed IOS (TRIOS; 3Shape) employed in this investigation did not require powder for the scanning procedure and permitted simple handling.

## 2.3. Group 2—Scanning the impression with the intraoral scanner

Initially, a cold-cured custom-made tray (Tray Material, Major, Moncalieri, Italy) was fabricated with three stops at the sides of the acrylic block (Fig 1A and 1C). A wax spacer was used to make a uniform 4-mm spacing to ensure optimal material thickness in the custom tray. Increasing the retention of the impression material was performed by drilling holes into each side of the tray. An adhesive (3M™ VPS Tray Adhesive, 3M Deutschland GmbH, Germany) was applied to the tray and kept dry over the counter for 12 min per the manufacturer's recommendation. Then, a conventional impression of the prepared maxillary molar was made with polyvinyl siloxane material (Bonascan; DMP Dental Industry S.A.), using heavy and light body consistencies through a two-step impression technique. The impression tray with the impression material inside was placed on the tooth model perpendicular to its long axis. It was held under finger pressure until the impression material was set per the manufacturer's recommendations and then lifted in a single motion. The impression-taking procedure was performed at room temperature, and the clinical setting was simulated by disinfecting the obtained impression (Impresept, 3M ESPE) for 10 minutes. To optimize the visibility of the proximal and occlusal tooth surfaces, the peripheral areas of the impression were trimmed using a scalpel. Before scanning, the impression was stored for 1 hour at room temperature and then digitized using the same IOS [38]. Without pouring the impression, ten scans were performed by the same operator (M.A.) [17].

## 2.4. Group 3—Scanning the impression with an extraoral scanner

Afterward, the impression was digitized using an extraoral laboratory scanner (3Shape D810; 3Shape, Copenhagen, Denmark). The same operator (M.A.) repeated the digitization method ten times without pouring the impression.

## 2.5. Group 4—Scanning the cast with the extraoral scanner

After 8 hours of storage, the obtained impression was poured using a type IV dental stone (Fujirock; GC Europe N.V.). According to the manufacturer's instructions, the impression was removed from the cast 45 minutes later. Following trimming and storage for 48 hours, the master cast was scanned with the same lab EOS. The scanning procedure was repeated ten times by the same operator (M.A.). Optical scans were stored as ten standard tessellation language (STL) files for each group.

## 2.6. Design and fabrication of endocrowns

A CAD software program (3Shape CAD Design software; 3Shape) was employed to design similar endocrowns on all virtual preparations, as demonstrated in Fig 2. For all samples, the cement spacer was fixed to be (a) 40 μm for the internal regions 1 mm below the margin and (b) no cement spacing (0 μm) for the margin itself [39]. The biogeneric reference option was used to design the occlusal anatomy for all endocrowns identically, ensuring that all restorations have the same design [40]. Following the designing step, a dental milling machine (DWX-50, Roland DG Corporation, Japan) was used to fabricate the wax patterns for future restorations. After fixing the wax blank (EASY blank wax, Renfert, Germany) in the milling chamber of the five-axis milling machine used in this study, the preview window was activated to begin milling the wax discs. An axial wax sprue (length: 5 mm, diameter: 3 mm) was fixed on the wax patterns along the material flow path. Each pattern was adjusted at an angle of 60º—the same practitioner (M.A.) designed and fabricated the wax patterns for standardization.

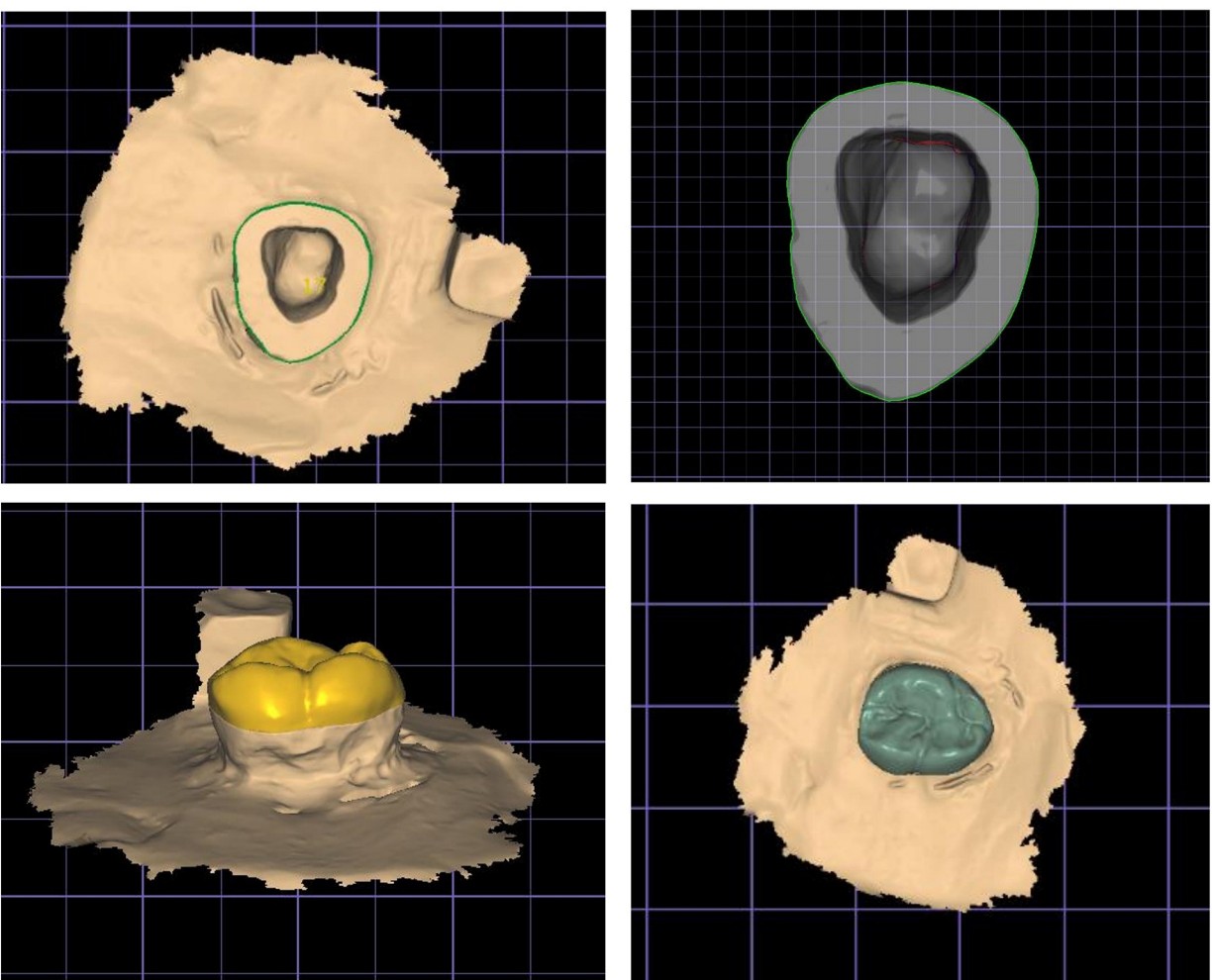

**Fig 2. Designing the endocrown with the laboratory CAD software.**

The wax patterns were invested using IPS PressVest (Ivoclar, Schaan, Liechtenstein) as a special investment material for IPS e.max. The investment ring was first pre-heated at 250˚C for half an hour without the ingot inside. Once further, the temperature elevated to 1100 degrees Celsius and held there for an hour. In this stage, a mold ready for pressing was obtained by removing the wax from the sprue channel. At 1100˚C, the placed ingot (IPS e.max Press, A2, low translucency, Ivoclar Vivadent) started plasticizing and was pressed at 3.5 bars inside the investment mold. After completing the program, the investment ring was removed from the furnace and cooled at room temperature for approximately 1 hour. A disc was then used to separate the ring. LDS fluorescent glaze paste (IPS Ivocolor glaze paste, Ivoclar Vivadent, Zürich, Switzerland) was employed to glaze the restorations. One skilled technician handled all laboratory procedures. The glazed endocrowns (Fig 1D) were examined for imperfections, including deformation, crack, or apparent marginal misfit, and were rejected in case of deficiency. The accuracy of marginal adaptation was confirmed in case the explorer could not enter between tooth margin and restorations, and two unbiased observers confirmed the passive fit. To assure the absence of internal interference with the proper seating of restorations, a fit checker spray (Arti-Spray, BK 285, Dr. Jean Bausch, Cologne, Germany) was employed to examine the IF of

all endocrowns on the tooth model. The faulty restorations underwent no adjustments, and the procedures were repeated to prepare and substitute the rejected specimens.

## 2.7. Measurement procedures

Each endocrown was seated on the tooth and held under finger pressure (Fig 1E). Images were taken with a digital camera attached to a microscope (AM413Fit Dino-Lite Pro; Dino-Lite electronic crop, Taipei, Taiwan). After connecting this microscope to a PC, the images were taken at a magnification of 80× (Fig 3). The measurement software was first calibrated by taking a digital photograph from a definite caliper distance at the same magnification and periodically measuring the distance of the caliper. Eight points were highlighted on the tooth 3 mm below the margin to measure the vertical MG at 45-degree intervals. Following that, photographs were captured from the explicitly designated points for each endocrown sitting. These pictures were analyzed using image analysis software (Dino Capture 2.0, AnMo Electronics Crop, Tainan Hsien, Taiwan). Using literature, the number of measures/samples was calculated [41,42]. This method made eight measurements on each sample, totaling 320

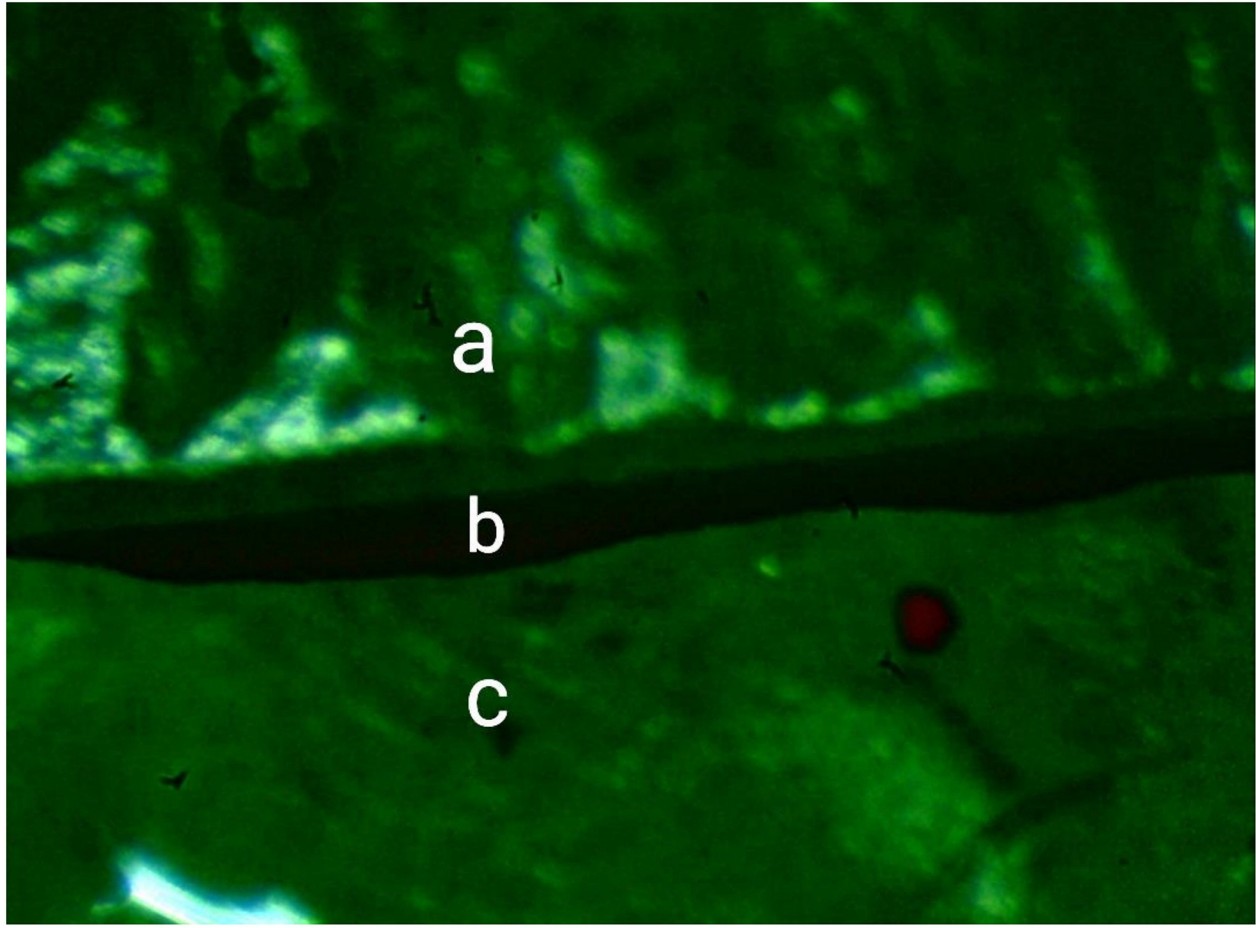

**Fig 3. Microscope image of the restoration-tooth interface at 80× magnification. (a)** Lithium disilicate endocrown; **(b)** Vertical marginal gap; **(c)** Prepared tooth.

measurements across four groups (4 groups×10 samples×8 measurements). The MG of every point represented the vertical marginal misfit (μm).

The IF and horizontal MF of the endocrowns were evaluated using the replica technique. The intaglio surface of each endocrown, coated with a light-body polyvinyl siloxane (Elite HD +; Light body-normal set, Zhermack, Italy), was placed on the tooth model mounted on a holding device. Following setting the material for 5 minutes and delicate removal of the restoration, the thin light-body layer adhered to the tooth surface was stabilized by adding a layer of regular-body material (Elite HD+; Tray material Regular body-normal set, Zhermack, Italy). Both the attached layers were removed from the model after the complete setting. Each replica was divided from the middle into four pieces along the buccolingual and mesiodistal directions. A fragment with 2 mm thickness and parallel walls was sliced from each sample to acquire a direct view from the stereomicroscope (Fig 4). Similar segmentations and sectioning slices were applied to each replica, enabling a perpendicular viewing from the stereomicroscope platform [33].

In line with the literature, each slice was divided into four areas, including axial wall (A), pulpal floor (P), marginal (M), and cervical (C) regions [40,43]. According to Fig 5, each section underwent eight measurements: marginal (n = 1; M1), cervical (n = 2; C1 at the center and C2 on the cervical-axial angle), axial (n = 3; A1-A3, which split the axial wall into three equally sized sections), and pulpal (n = 2; P1 at the axiopulpal angle and P2 at the center of the pulpal zone). M1 was defined as the distance between the external marginal line of the prepared model and the most extended point of the endocrown margin, C2 as the bisector of the angle between the axial wall and cervical area, P1 as the bisector of the angle between the axial wall and the pulpal floor and A1-A3, C1, and P2 as the perpendicular distances between the abutment tooth and the inner surface of the endocrown. M1 also denoted the horizontal MG and A1-A3, C1, C2, P1, and P2, the IF of the endocrown. This method used 1280 measurements across four groups, with 32 measurements on each sample. (4 groups×10 samples×4 pieces×8 measurements).

A prosthodontist (AA.K.) blinded to the study's design performed all measures. The same operator repeated all measurements after two weeks to ensure data reliability. The mean of two measurements was used for analysis. All the data was stored as an Excel file. Analysis was performed on the discrepancy thickness at internal and marginal areas zones [33]. The statistician who performed the analyses was also blinded to the study grouping. Fig 6 depicts the entire experimental procedure as a flow diagram.

## 2.8. Statistical analysis

SPSS software (IBM SPSS Statistics 24, IBM SPSS Inc., Chicago, USA) was used to statistically analyze the data. The Shapiro-Wilk test was performed to assess the normality of the data, and the Kruskal-Wallis test was used to compare the obtained values by the effect of studied digitization methods on the MG and IG of LDS endocrowns. The statistical significance was adjusted at P<0.05, and the intraclass correlation coefficient (ICC) was calculated to analyze the correlation between the two measurements.

## 3. Results

For the primary and secondary measurements over two weeks, the intra-observer agreement was deemed satisfactory (ICC = 0.878 to 0.926).

Table 1 presents the results obtained from the descriptive analysis of vertical MG, and Fig 7 shows the plot of each group. The Shapiro-Wilk test failed to meet the assumption of normality (P<0.05), and significant differences between the four groups were revealed by the Kruskal-

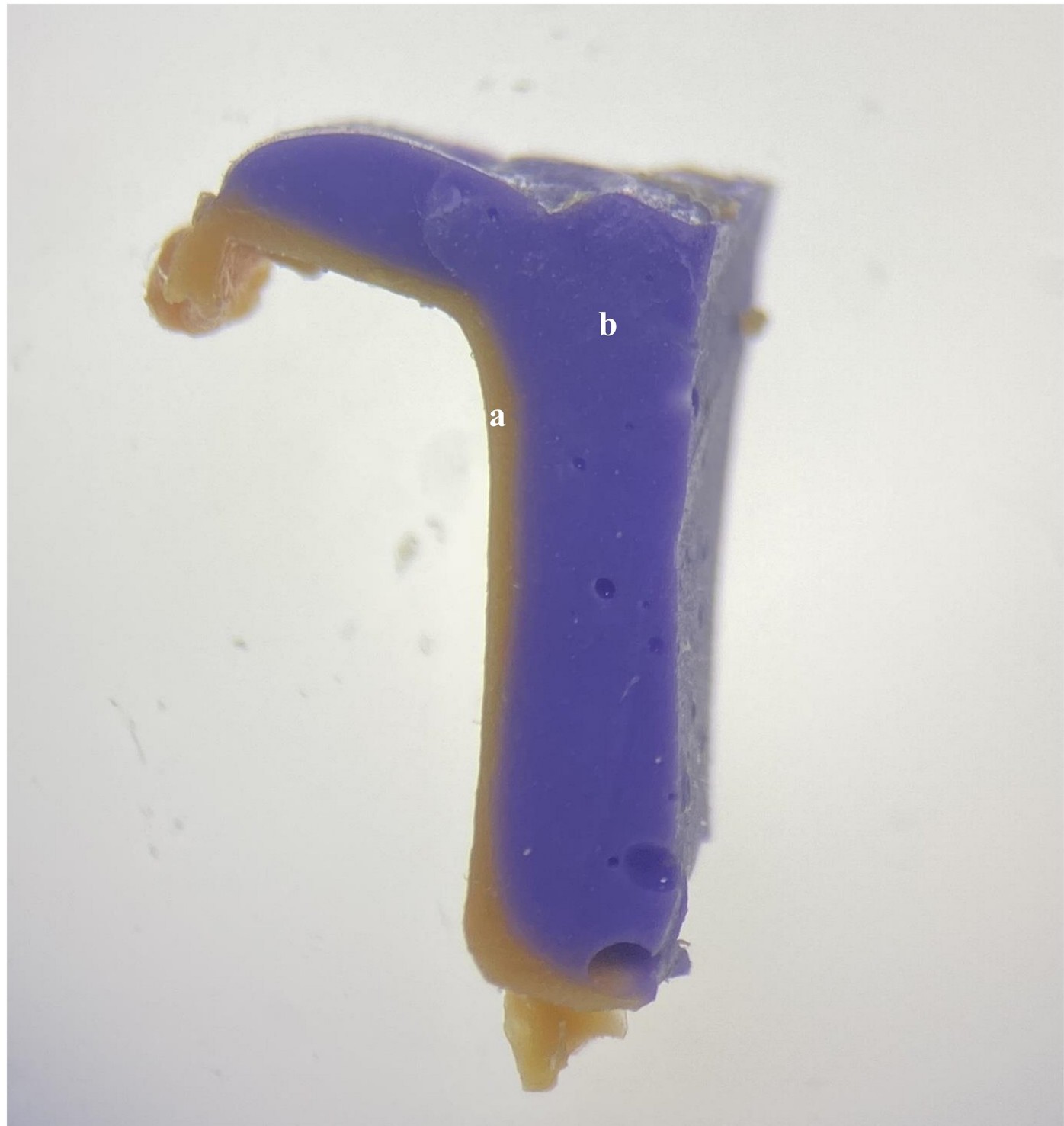

**Fig 4. Microscope image of the replica technique. (a)** Light-body layer representing internal adaptation; **(b)** Regular-body layer.

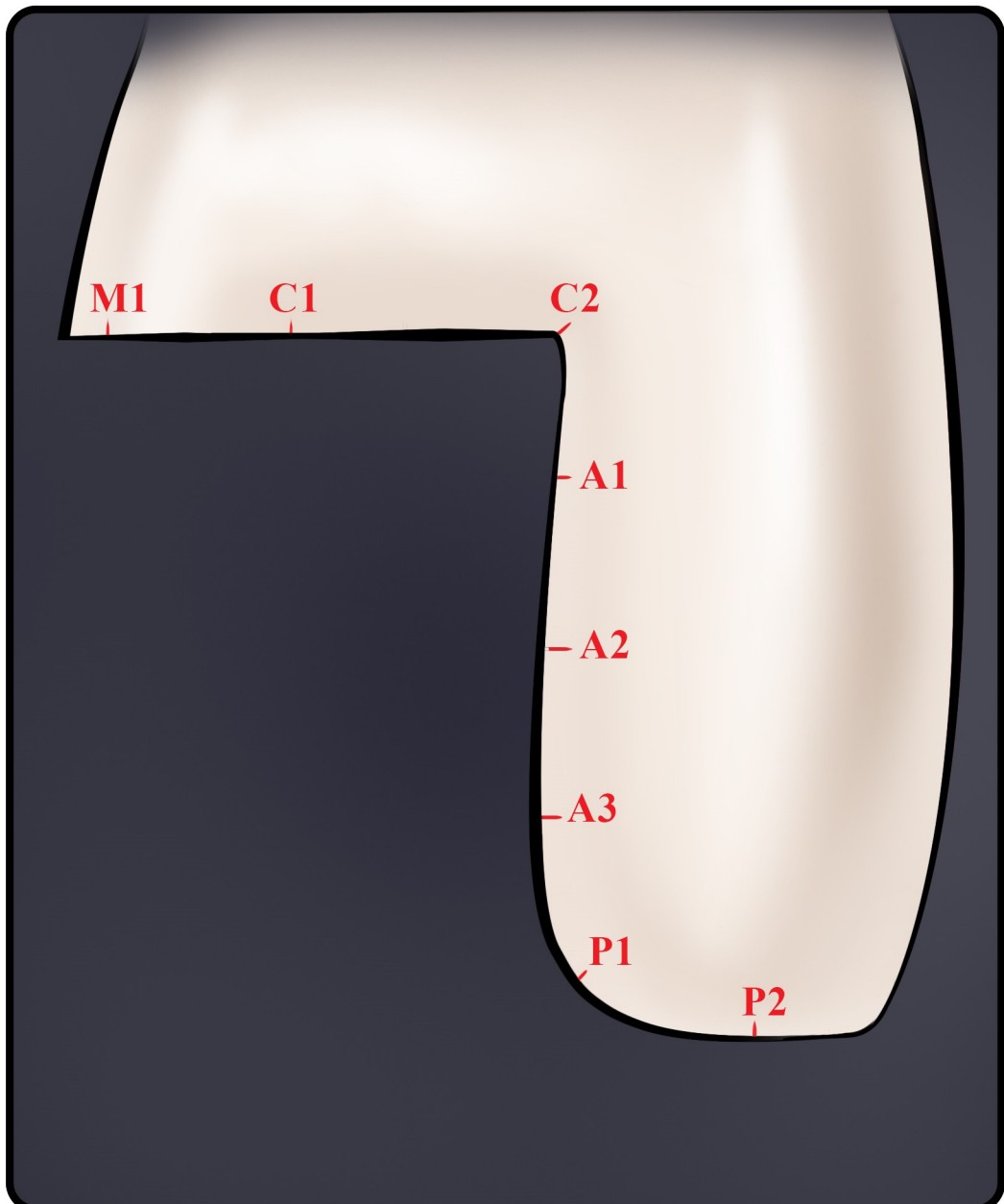

**Fig 5. A schematic illustration of the internal fit measurement points in the replica technique. P1** and **P2**: Pulpal discrepancies; **A1**, **A2**, and **A3**: Axial discrepancies; **C1** and **C2**: Cervical discrepancies; **M1**: Horizontal marginal discrepancy.

Wallis test (P<0.001). Scanning the impression by the IOS (G2) showed the most significant mean vertical MG (130.31±7.87 μm). Scanning of the impression by the EOS (G3) showed the smallest mean vertical MG (48.43±19.14 μm).

The descriptive analysis results regarding the marginal, cervical, axial, and pulpal regions obtained from the replica technique are demonstrated in Table 2. The Shapiro-Wilk test failed to support the data's normal distribution (P<0.05), and the Kruskal-Wallis test demonstrated

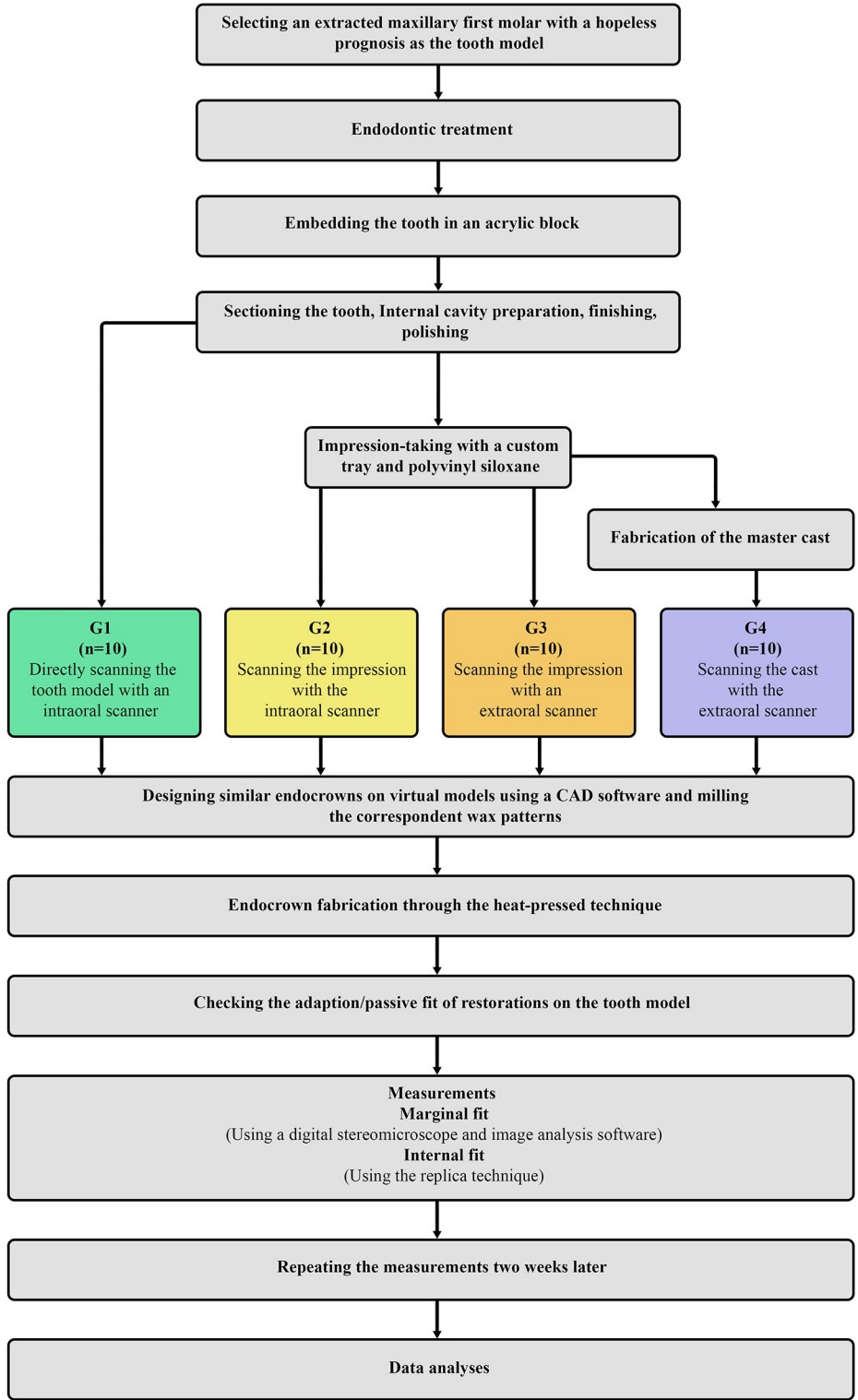

**Fig 6. A flowchart outlining the experimental procedures used to design this research.**

**Table 1. The average values of the endocrowns' vertical marginal gap (µm).**

| Study Groups | N* | Mean±SD† | Median | Minimum | Maximum |
|---|---|---|---|---|---|
| G1<br>Intraoral Scanner—Tooth Scan | 10 | 87.09±4.97 | 86.89 | 80.38 | 95.58 |
| G2<br>Intraoral Scanner—Impression Scan | 10 | 130.31±7.87 | 132.20 | 113.81 | 142.37 |
| G3<br>Extraoral Scanner—Impression Scan | 10 | 48.43±19.14 | 44.13 | 27.40 | 120.55 |
| G4<br>Extraoral Scanner—Cast Scan | 10 | 65.91±27.06 | 64.11 | 11.84 | 155.11 |

N*: Number; SD†: Standard Deviation.

significant differences among the four groups (P<0.001). Scanning the impression by the IOS (G2) showed the largest IG in all the four mentioned regions. In contrast, scanning the impression by the EOS (G3) directed the smallest IG in all four internal areas. The Kruskal-Wallis test demonstrated significant differences (P<0.001) among the four internal regions (horizontal

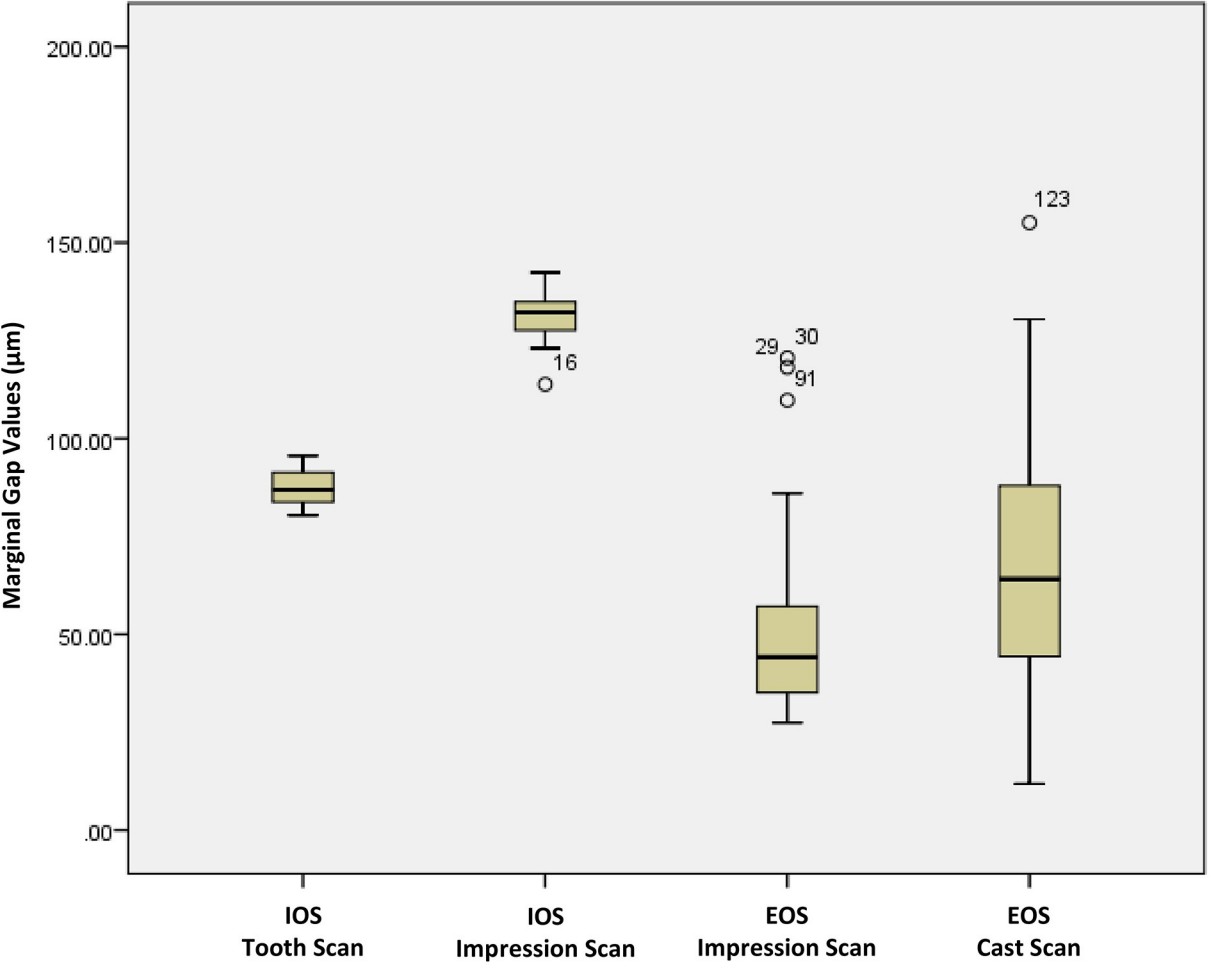

**Fig 7. The box-plot diagram of the vertical marginal gap (µm) comparisons among the four studied groups.**

**Table 2. Analysis of the gap in horizontal marginal, cervical, axial, and pulpal regions (µm).**

| Study Groups | Study Regions | Mean±SD† | Median | Minimum | Maximum |
|---|---|---|---|---|---|
| G1<br>Intraoral Scanner—Tooth Scan<br>(N* = 10) | Horizontal Marginal | 93.30±3.88 | 94.07 | 85.24 | 98.73 |
| | Cervical | 83.60±2.76 | 82.57 | 80.37 | 89.31 |
| | Axial | 78.77±3.21 | 79.05 | 74.46 | 85.64 |
| | Pulpal | 121.87±7.78 | 121.93 | 105.85 | 133.57 |
| G2<br>Intraoral Scanner—Impression Scan<br>(N* = 10) | Horizontal Marginal | 138.87±8.05 | 135.88 | 127.35 | 150.76 |
| | Cervical | 131.40±5.34 | 131.57 | 122.72 | 138.68 |
| | Axial | 128.07±3.28 | 127.32 | 124.19 | 134.80 |
| | Pulpal | 142.79±3.66 | 141.80 | 139.51 | 152.56 |
| G3<br>Extraoral Scanner—Impression Scan<br>(N* = 10) | Horizontal Marginal | 19.11±4.90 | 19.89 | 11.05 | 26.80 |
| | Cervical | 44.12±7.49 | 42.45 | 32.90 | 57.65 |
| | Axial | 73.35±10.57 | 70.96 | 53.07 | 89.83 |
| | Pulpal | 111.97±17.20 | 114.65 | 85.27 | 136.38 |
| G4<br>Extraoral Scanner—Cast Scan<br>(N* = 10) | Horizontal marginal | 31.13±5.18 | 31.29 | 23.31 | 43.20 |
| | Cervical | 58.86±6.61 | 59.86 | 46.07 | 67.88 |
| | Axial | 87.44±10.75 | 89.45 | 67.19 | 101.29 |
| | Pulpal | 95.67±15.68 | 92.50 | 68.88 | 119.54 |

N*: Number; SD†: Standard Deviation.

marginal, cervical, axial, and pulpal) studied within each group. The pulpal area showed the highest gap of the four measured regions (Table 2).

The results of the descriptive analysis of total IG (The mean calculated from cervical, axial, and pulpal values) are demonstrated in Table 3. The Shapiro-Wilk test failed to support the data's normal distribution ($P < 0.05$), and the Kruskal-Wallis test demonstrated significant differences among the four groups ($P < 0.001$). Scanning the impression by the IOS (G2) showed the largest total IG (133.56±3.58 µm). Scanning the impression by the EOS (G3) showed the smallest total IG (68.91±8.07 µm).

## 4. Discussion

The two main digital data acquisition techniques for fabricating ceramic restorations, including endocrowns, are indirect and direct scanning. A scannable elastomeric impression or a stone cast model is scanned in the indirect mode. In direct intraoral digitization, the

**Table 3. The average values of the fabricated endocrowns' total internal gap (µm).**

| Study Groups | N* | Mean±SD† | Median | Minimum | Maximum |
|---|---|---|---|---|---|
| G1<br>Intraoral Scanner—Tooth Scan | 10 | 92.57±2.12 | 92.29 | 90.22 | 97.41 |
| G2<br>Intraoral Scanner—Impression Scan | 10 | 133.56±3.58 | 133.22 | 129.81 | 139.98 |
| G3<br>Extraoral Scanner—Impression Scan | 10 | 68.91 ±8.07 | 71.51 | 56.61 | 78.83 |
| G4<br>Extraoral Scanner—Cast Scan | 10 | 75.32±7.87 | 78.40 | 59.19 | 82.18 |

N*: Number; SD†: Standard Deviation.

preparations are directly scanned without fabricating any impression or stone model [44,45]. The present study aimed to compare the IF and vertical MF of LDS endocrowns fabricated by different direct/indirect scanning protocols. To the authors' knowledge, this study is the first to compare these digitization methods using both IOS and EOS for fabricating LDS endocrowns. Based on the results, the four investigated groups in this study differed significantly regarding MF and IF; indirect digitization of the impression using EOS had significantly better MF and IF than other methods. Therefore, the null hypothesis was rejected.

LDS was chosen as the restorative material due to its superior mechanical attributes, including its high fracture toughness of 2–3 MPa, high flexural strength of 360–440 MPa, high resistance to thermal shocks, low thermal expansion, and crystals with minimal microcrack propagation, as well as its aesthetically pleasing qualities and bonding possibility. These features have turned LDS into the golden standard material among all glass-based ceramic restorations for the fabrication of endocrowns [46,47]. In this study, the heat-pressed method was employed to fabricate the restorations, given that surface details of the axio-pulpal line angles and pulpal floor irregularities, such as canal orifices and remaining gutta-percha, can cause restoration overmilling in the CAD/CAM workflow due to limitations in the size of milling instruments; therefore, it could cause flat surfaces such as pulpal cavity for bonding endocrowns no to be milled precisely, which yields a higher internal gap [9,48]. As a critical step with potential effects on the IF of endocrown restorations, the wax patterns were fabricated utilizing CAD/CAM technology to minimize human error and regulate any other potentially disruptive factors, such as the distortion of the wax pattern seen frequently in conventional methods [49,50]. Furthermore, scanning was conducted on a single tooth model for G1, a single scannable impression for G2 and G3, and a single master cast for G4 to reduce the potential for error and compare only the impacts of the scanning devices and techniques. The present research did not cement the endocrowns on the prepared tooth. The MF evaluation was therefore protected against cementation-associated factors such as luting agent viscosity, thickness, type, and seating forces while cementing restorations [51,52].

Clinically-acceptable MG and IG are not unanimously addressed in the literature. Different values within the range of 500–200 μm have been recommended for IG, horizontal, and vertical MG by various investigations to be clinically acceptable [53–57]. As a result, all values in the current experiment were regarded as within the permissible clinical range. The replica method and direct microscopic view, as non-destructive measurement techniques, were used to quantitatively analyze the internal and marginal accuracy, allowing for the reproducibility of assessments at various time intervals while leaving the tooth intact and evaluation of marginal adaptation at either predetermined points or the restoration's whole circumference [58,59].

Results showed that the scanning approach had an intervening influence on the gap thickness that was statistically significant. Using 3Shape D810 EOS to indirectly digitalize the taken impression (G3) showed the best results for all misfit types. Several studies have evaluated the accuracy and precision of IOSs and EOSs to date, with results that have been quite erratic, suggesting that EOSs offer nearly comparable or greater precision [17,60–65]. The present study employed the IOS in an extraoral setting, which allowed to compare the inherent accuracy of the two scanner types. However, in a clinical context, the patient's intraoral factors may be the potential cause of the disparity [66]. The 3Shape TRIOS 3 IOS was based on confocal laser technology using ultrafast optical sectioning that combines the projection of structured light and confocal microscopy. When scanning severely destructed teeth with limited geometric structure, such techniques are vulnerable to deviation pattern propagation [31]. EOSs, on the other hand, benefit from having several cameras and, more significantly, multi-axis motion movements, enabling more accuracy in scanning conventional impressions/casts. The 3Shape D810 EOS acts as an active triangulation device, emitting light of various wavelengths in strip patterns reflected

by structure surfaces and captured by a charge-coupled device. The superiority of the EOS mechanism can thus be used to explain the results [62]. Research also suggests the higher precision of EOSs in capturing undercuts and zones with high curvatures [60]. Besides, the D810 EOS line scanner technology with blue light has yielded superior results in the literature compared to other devices [67,68]. Blue light scanners use light sources with shorter wavelengths, reducing scanning errors for factors affecting the color and morphology of the scanned structure [67]. While IOSs employ the best-fitting algorithm to stitch the scanned images, EOSs may efficiently reconstruct the object's shape based on the point cloud produced in the three-dimensional point coordinate system [65]. EOSs automatically scan a fixed model at various angles to lower operator-associated factors, potentially influencing the scanning accuracy [69]. Contrary to our findings, a few earlier investigations found statistically insignificant differences between intraoral and extraoral digitization approaches regarding the MF of restorations [25,70,71]. However, these studies mainly focused on scanning the gypsum die with EOS/IOS rather than digitization of the taken impression. Güth et al. [72] found that the accuracy of ascertained datasets depends on the scanning system, and direct digitization does not outperform indirect digitization in any studied systems. Another study found the MF of 4-unit zirconia fixed dental prostheses (FDPs) after direct scanning to be comparable to indirect digitization [13] despite the higher accuracy of the datasets produced by the direct digital approach [73]. This indicates that the higher accuracy of the virtual model datasets did not translate into a better MF of the final restorations [74]. Abduljawad and Rayyan [7] found the endocrowns fabricated by directly digitalizing the tooth using an IOS or cast digitization with IOS or EOS to yield statistically insignificant differences in the mean MG and other discrepancies at cavity walls, pulpal floor, and line angles. This dissimilarity with our findings could be related to the different study designs, measuring techniques, and employing multiple tooth samples in each group, making the standardization approach challenging. Ahrberg et al. [44] found a substantially superior MF employing IOS in the fabrication of restorations. However, they only studied scanned casts and not the impressions; they used different IOS and EOS systems compared to our study, and the material tested was zirconia frameworks veneered with LDS. The potentially-significant changes during the veneering [75] and sintering of pre-sintered zirconia [74] could affect the final dimension and density of the restoration and increase the MF.

Comparing the IOS techniques, directly scanning the tooth showed better adaptation than scanning the impression in all regions. The interval between taking the impression and scanning may introduce an extra distortion, leading to higher MG and IG [76]. Besides, the scanning of the tooth and impression, as different surfaces, have various degrees of accuracy due to their distinct optical characteristics [77]. However, it must be considered that in-vivo studies may lead to different results due to the presence of patient-related factors influencing the intraoral scanning procedure, such as subgingival finish lines, intraoral moisture (saliva or blood), the presence of soft tissue, the opposing dentition and neighboring teeth, movements of the patient and limited space of the oral cavity, especially in the molar area. Additionally, the majority of IOSs construct digital models using image-stitching techniques. While scanning, the handheld IOS device oscillates, necessitating frequent coordinate adjustments [78]. Thus, processing and matching errors accumulate with each stitching of the acquired image, leading to image distortion and erroneous readings [79]. The higher the complexity and geometry of the remaining tooth structure/preparation, the lower the accuracy of IOS in data acquisition [62,69,72,80].

Studies have also demonstrated high precision of the impression scanning technique with EOSs [68,81–84]. The reason that impression scanning by EOS gave better results compared to cast scanning could be attributed to the rougher surface of the casts and additional laboratory steps involved in producing gypsum casts, including inherent errors associated with pouring

the master cast, the expansion of the dental stone, trimming effects, and time-associated cast deformation [29,85]. Scanning the impression thus eliminated error-prone extra laboratory stages. Using a similar study design, Runkel et al. [86] showed that digitizing the impression without the additional step of master cast fabrication appeared to be a practical approach for scanning short spans up to one quadrant for the following restoration fabrication processes. A similar study methodology was used by Akhlaghian et al. to assess the MF of zirconia copings made utilizing direct and indirect digitization with EOS and IOS. They came to the same conclusion as our investigation: scanning the impression with a laboratory scanner was the most effective digitization approach, producing copings with the lowest MG [17].

The pulpal floor in the current study most significantly displayed the highest gap among all evaluated groups (P<0.05). In line with the present research, an investigation by El Ghoul et al. on the MF and IF of LDS endocrowns reported the most prominent gap at the pulpal floor of all the study groups [33]. This finding can be explained by overshooting near the edges and the imperfect scanning of the pulpal floor due to the limited optical depth of scanners for scanning the tooth and gypsum casts [40,43,87]. The significant differences can also be related to the surface anatomy, with the more uneven surface of the pulpal floor than that of the axial wall and the distance between the scanner tip and a deep pulp chamber, increasing the IG in the digitization of the tooth and poured casts. The restricted optical depth of the device causes blurred images from the unsmooth pulpal floor and axio-pulpal line angles, leading to lower accuracy and a higher gap in these regions [37,48]. On the other hand, in G2 and G3, as the samples were negative records of the prepared model, the pulpal floor scan could be expected to lead to restorations with better adaptability in this region. However, this outcome might have been negatively affected by the maximal ceramic thickness at the pulpal area, which experienced more significant dimensional distortion through the heat treatment process [55]. In this study, heat-pressed method was employed to fabricate the restorations given that surface details of the axio-pulpal line angles and pulpal floor irregularities, such as canal orifices and remaining gutta-percha, can cause restoration overmilling in the CAD/CAM workflow due to limitations in the size of milling instruments; therefore, it could cause flat surfaces such as pulpal cavity for bonding endocrowns no to be milled precisely, which yields a higher internal gap [9,48].

A direct comparison between different studies is limited due to the lack of a standardized protocol regarding the studied restoration types (crowns, FPDs, inlays, onlays, and endocrowns), the accuracy of the scanning systems, the cement space, preparation designs, materials used, fabrication techniques and measurement methods. The results of this study might have been influenced by the in-vitro nature of the study design, which was not similar to clinical settings with the presence of additional factors such as subgingival finish lines, intraoral moisture (saliva or blood), the presence of soft tissue, movements of the patient and limited space of the oral cavity, especially in the molar area. The shrinkage of polyvinyl siloxane following thermal changes when removed from the oral cavity was not considered, as the impression in this study was taken in an extraoral setting. Moreover, the gaps were measured without endocrowns cementation on the prepared teeth, leading to variations in the measurement values. Thus, more realistic conclusions could be drawn from further research with an in-vivo design and a larger sample size to involve the mentioned intraoral considerations in a clinical setting.

## 5. Conclusion

The present in-vitro study is concluded as follows:

1. The marginal adaptation of all lithium disilicate endocrowns fabricated following the digitization of the master model by four digitization methods, using intra and extraoral scanners, was found to be clinically acceptable.

2. The impact of digitization techniques on marginal and internal adaptation is evident, with indirect digitization of acquired impressions through an extraoral scanner demonstrating notably reduced marginal and internal gaps when compared to alternative methodologies.

3. The gap was maximized at the pulpal floor of all the study groups.

4. Adopting a hybrid workflow to digitalize a conventional impression can help avoid the time-consuming fabrication of casts, omit superfluous stages in the working process, and allows for the capture of fine details, ensuring high accuracy in the reproduction of dental structures by lowering the intervening errors. By harnessing the positive features of both conventional and digital technologies, the hybrid approach also covers a wider variety of cases. It is especially helpful in challenging situations when it could be difficult to digitally capture certain details. When dentists are unable to use intraoral scanners due to financial constraints, the hybrid process is also advantageous. In this scenario, traditional impressions using silicone materials are initially obtained, then the laboratory scans and digitizes the impressions to provide a precise and effective digital workflow at a reasonable cost.

## Supporting information

**S1 File.**
(XLSX)

## Acknowledgments

The authors thank Dr. Kamran Mirzaei from the Dental Research Development Center for the statistical analysis.

## Author Contributions

**Conceptualization:** Marzieh Akhlaghian.

**Data curation:** Marzieh Akhlaghian, Sana Dabiri, Farhad Kadkhodae, Shabnam Gholami.

**Formal analysis:** Sana Dabiri, Farhad Kadkhodae, Shabnam Gholami.

**Funding acquisition:** Marzieh Akhlaghian.

**Investigation:** Marzieh Akhlaghian, Amir-Alireza Khaledi, Seyed Ali Mosaddad, Sana Dabiri, Farhad Kadkhodae.

**Methodology:** Marzieh Akhlaghian, Amir-Alireza Khaledi.

**Project administration:** Marzieh Akhlaghian.

**Resources:** Marzieh Akhlaghian.

**Software:** Sana Dabiri, Farhad Kadkhodae, Shabnam Gholami.

**Supervision:** Marzieh Akhlaghian, Amir-Alireza Khaledi, Rashin Giti.

**Validation:** Marzieh Akhlaghian.

**Visualization:** Amir-Alireza Khaledi, Seyed Ali Mosaddad, Rashin Giti.

**Writing – original draft:** Seyed Ali Mosaddad.

**Writing – review & editing:** Seyed Ali Mosaddad.

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
