## [Decision Letter · Decision Letter 0]

30 Jan 2024

PONE-D-23-40785The internal and marginal adaptation of lithium disilicate endocrowns fabricated using intra and extraoral scanners: An in-vitro studyPLOS ONE

Dear Dr. Giti,

Thank you for submitting your manuscript to PLOS ONE. After careful consideration, we feel that it has merit but does not fully meet PLOS ONE’s publication criteria as it currently stands. Therefore, we invite you to submit a revised version of the manuscript that addresses the points raised during the review process.

 Please address the concerns raised by the reviewers and revise the manuscript accordingly. I look forward to receiving your revised submission.

We look forward to receiving your revised manuscript.

Kind regards,

Mirza Rustum Baig

Academic Editor

PLOS ONE

Journal Requirements:

2. In the online submission form, you indicated that [The data underlying the results presented in the study are available from the corresponding author.]. 

Reviewers' comments:

Reviewer's Responses to Questions

**Comments to the Author**

1. Is the manuscript technically sound, and do the data support the conclusions?

Reviewer #1: Yes

Reviewer #2: Yes

2. Has the statistical analysis been performed appropriately and rigorously? 

Reviewer #1: Yes

Reviewer #2: Yes

3. Have the authors made all data underlying the findings in their manuscript fully available?

Reviewer #1: Yes

Reviewer #2: Yes

4. Is the manuscript presented in an intelligible fashion and written in standard English?

Reviewer #1: Yes

Reviewer #2: Yes

5. Review Comments to the Author

Reviewer #1: The paper is well written. few formatting issues like titles of figures and tables. more important, you need to justify why you did not mill the restoration directly. why did you use the wax pattern to fabricate the restorations?

Reviewer #2: Briefly mention the limitations of the in-vitro setting and the need for further in-vivo research.

Consider quantifying the "significantly smaller" gap achieved with the optimal method for additional clarity.

You could optionally elaborate on the potential benefits of the proposed hybrid workflow beyond reducing errors.

Overall, this conclusion effectively communicates the study's contribution to the field and paves the way for further research and development.

6. PLOS authors have the option to publish the peer review history of their article (what does this mean?). If published, this will include your full peer review and any attached files.

Reviewer #1: No

Reviewer #2: No

---

## [Author Response · Author response to Decision Letter 0]

2 Feb 2024

Dear Editor and Reviewers,

We appreciate your precious time in reviewing our paper and providing valuable comments. Your insightful words led to possible improvements in the current version. We have carefully considered the statements and tried our best to address every one of them. Besides, the entire text has been thoroughly revised to rectify English writing mistakes. We hope the manuscript, after careful revisions, meets your high standards. The authors welcome further constructive comments, if any. We provide point-by-point responses as follows.

Reviewer #1: The paper is well written. few formatting issues like titles of figures and tables. more important, you need to justify why you did not mill the restoration directly. why did you use the wax pattern to fabricate the restorations?

RESPONSE: Thank you for your valuable comment. We have modified the in-text citations of the figures to accord with the journal's guidelines. Furthermore, we have revised the figure and table caption to ensure clarity and correctness. Regarding the fabrication method, we have added to the discussion the reason we opted for the heat-pressed technique rather than milling restoration and why we used milling the wax pattern in the heat-pressed technique rather than going through the conventional wax-up process. 

Reviewer #2: Briefly mention the limitations of the in-vitro setting and the need for further in-vivo research.

RESPONSE: We have dedicated a separate section to the limitations of the in vitro setting and the justification of the need to conduct more clinical studies. Thank you.

Consider quantifying the "significantly smaller" gap achieved with the optimal method for additional clarity.

RESPONSE: We appreciate this observation. We have revised this statement in the background to ensure it has enough clarity for the readers. Since this is the conclusion section, we avoided providing numerical data to support our statement. However, we have added that this smaller gap was in comparison to other techniques employed in this study. Additionally, the statistical data is sufficiently provided in the result section and further discussed in the discussion section. 

You could optionally elaborate on the potential benefits of the proposed hybrid workflow beyond reducing errors.

RESPONSE: Thank you for this great feedback. We have expanded the corresponding section in the conclusion to state the benefits of the hybrid approach further.

Overall, this conclusion effectively communicates the study's contribution to the field and paves the way for further research and development.

RESPONSE: Thank you very much for your time and consideration in reviewing this manuscript.

---

## [Editor Report · Decision Letter 1]

1 Mar 2024

The internal and marginal adaptation of lithium disilicate endocrowns fabricated using intra and extraoral scanners: An in-vitro study

PONE-D-23-40785R1

Dear Dr. Giti,

We’re pleased to inform you that your manuscript has been judged scientifically suitable for publication and will be formally accepted for publication once it meets all outstanding technical requirements.

Kind regards,

Mirza Rustum Baig

Academic Editor

PLOS ONE
---

## [Editor Report · Acceptance letter]

4 Apr 2024

PONE-D-23-40785R1 

PLOS ONE

Dear Dr. Giti, 

I'm pleased to inform you that your manuscript has been deemed suitable for publication in PLOS ONE. Congratulations! Your manuscript is now being handed over to our production team.

Kind regards, 

on behalf of

Dr. Mirza Rustum Baig 

Academic Editor

PLOS ONE